# Lived experiences of postpartum hypertensive women in Accra, Ghana

Kennedy Dodam Konlan[1]*, Hellen Akosua Asante[1], Cecilia Eliason[1]

1 Department of Adult Health, School of Nursing and Midwifery, University of Ghana, Legon, Greater Accra Region, Ghana

* kennedy.konlan@gmail.com

## Abstract

### Background

Hypertension is a leading cause of adult mortality globally with postpartum hypertension posing distinct risk to women. In Ghana, particularly within the Accra Metropolis, there is a limited understanding of the lived experiences of women managing hypertension after childbirth.

### Aim

We described the lived experiences of postpartum women diagnosed with hypertension receiving care in a resource-constrained setting in Accra, Ghana.

### Methods

The study employed a qualitative, descriptive-phenomenological design. We purposively selected 16 postpartum women diagnosed with hypertension and receiving care. They were recruited from the postnatal clinic of the Greater Accra Regional Hospital. The data was collected via in-depth, semi-structured interviews using a pretested interview guide. The interviews were audio-taped and transcribed verbatim. We then conducted thematic analysis with the aid of NVivo 11.0.

### Results

The analysis yielded three major themes; expectations about living with hypertension, self-efficacy in managing hypertension and environmental factors on the management of hypertension. On the expectation about living with hypertension, we identified the following sub-themes: optimism and hope for recovery as well as fear and uncertainty about complications. Regarding self-efficacy, the following sub-themes were identified: Confidence in adhering to treatment and challenges with lifestyle modification and monitoring. On the environmental factors influencing management

**Data availability statement:** All relevant data are within the paper and its Supporting Information files.

**Funding:** The author(s) received no specific funding for this work.

**Competing interests:** The authors have declared that no competing interests exist.

**Abbreviations:** GARH, Greater Accra Regional Hospital; SCT, Social Cognitive Theory.

of hypertension among the participants, we found the following sub-themes, spousal and family support, role of healthcare providers, social perceptions and stigma.

## Conclusion and recommendations

Postpartum women diagnosed with hypertension face social stigma and have mixed feelings of hope, fear and uncertainty about developing complications. We recommend that midwives/nurses should design and implement targeted postpartum hypertension education programmes to promote confidence in the treatment, improve knowledge on the condition and enhance emotional resilience. Also, managers of antenatal and postnatal clinics must integrate spouses and faith-based organizations in the management of postpartum hypertension to reduce the associated social stigma.

## Introduction

Hypertensive disorders remain among the foremost causes of maternal morbidity and mortality globally [1–7], yet the postpartum period continues to receive limited attention despite its high-risk nature [4]. Postpartum hypertension, which may represent persistent or new-onset high blood pressure following childbirth, is increasingly recognized as a critical maternal health issue [2,4]. Recent studies [1–7] estimate that approximately 28.4% of women who experience hypertensive disorders of pregnancy continue to have hypertension within two years postpartum, compared with 9.1% of those without prior hypertensive disorders [8]. In low-resource settings, as many as 62.1% of women with preeclampsia or eclampsia remain hypertensive one year after delivery [1,9]. Moreover, even among women with uncomplicated pregnancies, about 3% develop de novo hypertension within the first postpartum year [4,10].

In Africa, the burden of postpartum hypertension is notably higher due to delayed diagnosis, limited access to quality prenatal and postnatal care, and inadequate treatment strategies [11–13]. For instance, research in Nigeria revealed that 12% of postpartum women developed hypertension, a situation worsened by weak health systems and cultural influences [13]. In Uganda, prevalence was reported at 14%, largely attributed to gaps in follow-up care [13] while in South Africa, around 10% of postpartum women were affected, with striking differences in healthcare access between rural and urban areas [4,13]. In Ghana, available data points to a rising trend, with estimates suggesting 10–15% of mothers experience postpartum hypertension, particularly in urban centers such as Accra [14]. Similar figures were found in other African countries: 11% in Kenya, where inadequate monitoring was a key contributor [13], and 13% in Ethiopia, where late diagnosis often led to life-threatening complications [4,15].

In Ghana, hypertensive disorders in pregnancy continue to impose a significant burden on maternal health. A recent study reported a prevalence rate of 37.2% among antenatal clinic attendees, with 17.6% presenting chronic hypertension superimposed on preeclampsia [16]. These statistics underscore the need for continued

postpartum follow-up, as unresolved or newly developed hypertension after delivery contributes to long-term cardiovascular, renal, and cerebrovascular complications [8,17]. Beyond its physical health implications, postpartum hypertension also exerts considerable psychological, social, and economic pressures on mothers, particularly in resource-constrained contexts. Many affected women struggle to adhere to antihypertensive medications while balancing newborn care, follow-up appointments, and limited social or family support [10,18]. The compounded stress and anxiety may further compromise health outcomes and family stability [17–19].

Several studies [1–12] have revealed that mothers often reflect on feelings of vulnerability and loss of control over their bodies during the postpartum period. Many women express surprise that hypertension did not resolve immediately after birth, which contradicts their expectations and leads to emotional distress [11–19]. These experiences are often compounded by insufficient communication with healthcare providers, leaving mothers uncertain about medication regimens and the need for ongoing monitoring. Consequently, a sense of mistrust and frustration may develop toward the healthcare system, as women feel their health concerns are overshadowed by the focus on newborn care [17,19]. Despite growing recognition of postpartum hypertension as a major maternal health concern, limited qualitative research have explored the lived experiences of postpartum women diagnosed with hypertension and receiving care in Ghana. This gap in understanding hampers the ability of healthcare professionals and policymakers to design holistic, woman-centered interventions that address both the clinical and psychosocial needs of postpartum women living with hypertension in resource-constrained settings like those in Ghana.

### Aim

We described the lived experiences of postpartum women diagnosed with hypertension receiving care in a resource-constrained setting in Accra, Ghana.

### Theoretical framework underpinning the study

Understanding the lived experiences of postpartum women with hypertension requires a multidimensional framework that explains how personal beliefs, environmental factors, and social influences shape health-related behaviors. Several behavioral theories, including the Health Belief Model (HBM), the Theory of Planned Behaviour (TPB), and the Social Cognitive Theory (SCT), provide relevant perspectives for understanding how women perceive and manage hypertension in the postpartum period. Among these frameworks, SCT provides the most comprehensive foundation for this study because it accommodates the interplay between individual, social, and environmental dimensions of experience. It thus enables a deeper understanding of how postpartum women living with hypertension in the Accra Metropolis perceive, learn, and act in ways that influence their coping and management strategies.

The Social Cognitive Theory (SCT), developed by Bandura, offers a more holistic approach by emphasizing the reciprocal interaction between personal factors, behavior, and environmental influences, a process known as reciprocal determinism [20,21]. Key constructs such as observational learning, self-efficacy, and social reinforcement make SCT particularly suitable for understanding how postpartum mothers adopt coping strategies to manage hypertension. For example, women may learn effective self-care behaviors through observing peers, receiving encouragement from family or community health workers, and developing confidence in their ability to control their condition. The SCT aligns well with Ghana's communal and family-centered culture, where social modelling and collective coping play critical roles in shaping health behavior. However, the broad scope of SCT can make empirical measurement of some constructs complex in qualitative research [21]. Albert Bandura's Social Cognitive Theory (SCT), introduced in the 1980s, provides a broad framework for explaining behavior through the interaction of personal factors, environmental influences, and behavior itself [20]. Its central principle, reciprocal determinism, posits that an individual's behavior is shaped not only by internal beliefs but also by social and environmental conditions. In maternal health, for example, adherence to hypertension management is influenced by a mother's confidence, family support, and healthcare access [20].

The Social Cognitive Theory emphasizes the dynamic interplay between personal attributes, behaviors, and environmental conditions in influencing human actions [20,21]. Applied to postpartum hypertension, SCT offers a comprehensive framework for understanding how mothers interpret and manage their condition through constructs such as self-efficacy, outcome expectations, observational learning, and social support. This perspective is particularly useful in explaining how women balance the physical, emotional, and social challenges of the postpartum period while living with hypertension.

Key constructs of the SCT include:

Self-efficacy: the belief in one's ability to perform health behaviors, essential for medication adherence and follow-up.

Observational learning: adopting behaviors by observing others, e.g., attending ANC after witnessing complications in peers.

Outcome expectations: beliefs about the benefits of a behavior, such as reduced complications through regular monitoring.

Behavioral capability: knowledge and skills necessary for performing health behaviors, which can be strengthened through health education.

Reinforcements: external or internal rewards that sustain behavior, e.g., encouragement from health workers.

Self-regulation: setting goals, monitoring progress, and adjusting behaviors to manage health, such as dietary changes or blood pressure checks.

In pregnancy and postpartum hypertension, SCT emphasizes how cultural norms, healthcare structures, and family support shape women's ability to manage their condition. Interventions based on SCT encourage building confidence, strengthening family/community involvement, and reinforcing positive health behaviours. While theory is comprehensive, it poses challenges, such as the difficulty of measuring certain constructs (e.g., self-efficacy) and the need for longitudinal designs to confirm causality. Nevertheless, SCT (Fig 1) remains highly valuable for designing culturally appropriate, multi-dimensional interventions for maternal hypertension.

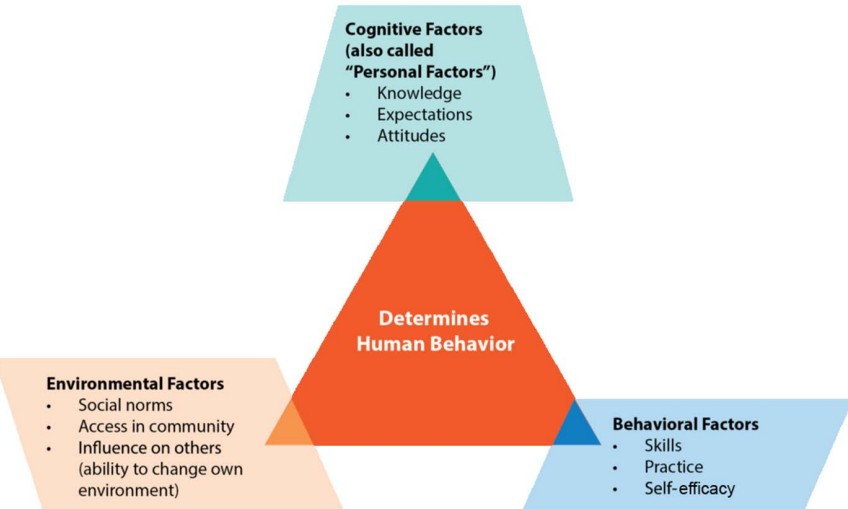

**Fig 1. Diagrammatic representation of the SCT adopted from [20].**

## Method

### Study design

The study employed a descriptive phenomenological approach to investigate the lived experiences of postpartum women diagnosed with hypertension in the Accra Metropolis. Grounded in Edmund Husserl's philosophical perspective, descriptive phenomenology seeks to uncover the essence of human experiences by suspending researcher assumptions through the process of bracketing. This method was chosen to ensure that women's voices were authentically represented, focusing on the "what" and "how" of their experiences without imposing researchers' interpretation.

### Study setting

The study was conducted at the Greater Accra Regional Hospital (GARH), commonly referred to as Ridge Hospital, situated at No. 1 Haile Selassie Street, North Ridge, Accra. The facility is centrally located, making it accessible from different parts of the city. The Greater Accra Region has a population of about 5,055,883, with females representing 50.7% of the national population [5,6]. It is one of the most densely populated regions, with an estimated 1,205 people per square kilometer. The region has an extensive network of public and private health facilities. GARH was selected as the study site because of its strategic role as a major referral center in the region and its high caseload of pregnant and postpartum women, particularly those with hypertensive disorders [17,19]. Data suggest that approximately 8.5% of pregnant women attending the facility experience hypertensive conditions [18]. This setting provided access to a diverse pool of participants and offered an opportunity to examine the phenomenon within a busy clinical environment.

### Target population and sample size determination

The study population consisted of all postpartum women aged 18 years to 45 years diagnosed with hypertension who were receiving postnatal care at GARH.

The sample size was determined at data saturation which was achieved at the conclusion of the sixteenth (16th) interview. We observed data redundancy after the conclusion of the twelfth interview and after concluding four additional interviews, we observed that no new information was being elicited hence data saturation at the 16th interview.

Inclusion criteria:

Women with a confirmed diagnosis of hypertension (≥140/90 mmHg) on at least three separate measurements while at rest.

Attending postnatal clinic at GARH.

Cognitively stable which was assessed as women who engaged in coherent communication during the interaction with members of the research team

Women aged 18–45 years. The minimum age limit was set due to the legal age for consent in Ghana while the maximum age limit was set so as to ensure women in fertile/reproductive age were engaged for the study.

Exclusion criteria:

Women too ill to participate.

Those with hearing or speech impairments preventing participation.

### Selection of participants and data collection

Participants were selected if they were diagnosed with hypertension during the postpartum period. This approach enabled us to identify individuals who could offer valuable insights into their experiences living with postpartum hypertension, thereby enriching the qualitative data for the study.

The data collection took place between 4th October 2025–30th October 2025. Ethical approval was obtained from the Ghana Health Service Ethics Review Committee (GHS – ERC: 028/06/25). An introductory letter from the School of Nursing, University of Ghana, was presented to the hospital to facilitate access. Nurses at the postnatal clinic were briefed on the study's objectives and eligibility criteria. Potential participants were identified by the research team, who then contacted them, explained the study in detail, answered their questions, and screened them for eligibility. Interested women were given information sheets and consent forms. Appointments were arranged at convenient times and locations for interviews. All the participants agreed to take part in the interviews by signing the informed consent forms. We conducted the interviews at a consulting room assigned to the research team at the postnatal clinic and the said room was only accessible to the researchers and the participants only during the period of data collection

The participants were interviewed using a semi-structured interview guide (S1). The interview guide consisted of two sections: the first captured demographic information, while the second focused on the participants' expectations and lived experiences related to living with postpartum hypertension. Each interview began with a broad, introductory ("grand tour") question to encourage open discussion, followed by targeted probes to explore specific areas of experience in greater depth. Field notes were also taken to capture non-verbal cues and contextual observations.

The interviews lasted between 30–45 minutes and were conducted in English (the official language of communication in Ghana) and two native Ghanaian languages widely spoken in Accra (Twi and Ga). The interviews were recorded using an audio recorder, transcribed verbatim after each interview. We further did member checking with the participants to ensure their exact views had been captured.

## Data management

To maintain confidentiality, pseudonyms were assigned to participants (e.g., names of the alphabets were used). Audio recordings were transcribed verbatim immediately after interviews. Files were stored with unique identifiers, and transcripts were separated from demographic details. Hard copies were kept securely, while digital files were password-protected on the researcher's computer.

## Methodological rigor

To strengthen the accuracy and dependability of the results, the researcher implemented measures consistent with the four recognized principles of trustworthiness, credibility, dependability, confirmability, and transferability as outlined in contemporary qualitative research standards.

Credibility was strengthened through prolonged engagement, member checking, and triangulation. Prolonged engagement involved building trust with mothers during clinic visits before interviews were conducted. Member checking allowed participants to confirm or clarify interpretations of their responses, ensuring accuracy of representation. In addition, peer debriefing with fellow researchers and supervisors was carried out to refine emerging themes, which increased the authenticity of findings.

Dependability was ensured through the creation of a detailed audit trail that documented each step of the research process, including interview procedures, field notes, transcription records, and analytical decisions. This systematic documentation provides transparency and allows other researchers to understand how the findings were derived. The researcher also maintained a reflective journal throughout the study to record decisions, challenges, and insights that emerged during data collection and analysis. This practice contributed to consistency and accountability in the research process.

Confirmability, which focuses on minimizing researcher bias and ensuring that findings are grounded in participants' perspectives rather than the researcher's assumptions, was achieved through reflexivity. The researcher acknowledged her professional background as a midwife and consciously bracketed personal experiences and preconceived ideas about postpartum hypertension. Regular self-reflection and adherence to the interview guide helped maintain neutrality during

data collection and analysis. Furthermore, the inclusion of verbatim quotes from participants in the findings ensured that interpretations were rooted in their authentic voices rather than subjective inference.

Transferability was promoted through detailed descriptions of the study context, participants' demographic details, and factors influencing postpartum hypertension. Although statistical generalization is not the aim of qualitative work, providing rich contextual detail allows readers to judge the applicability of findings to similar settings.

Finally, ethical rigour was ensured through informed consent, confidentiality, and securing ethical approval from the Ghana Health Service Ethics Review Committee, supported by an introductory letter from the School of Nursing, University of Ghana. Participants were reminded of their right to withdraw from study at any stage without any consequences. Collectively, these measures enhanced the trustworthiness of the inquiry.

Ethical approval and consent to participate.

Ethical clearance was obtained from the Ghana Health Service Ethics Review Committee (GHS – ERC:028/ 06/ 2025), supported by a letter from the School of Nursing, University of Ghana. Potential participants were given information sheets detailing the purpose, objectives, possible benefits, risks, and inconveniences associated with the study. Confidentiality, voluntary participation, and the right to withdraw at any point were emphasized. Interviews were conducted in English at venues preferred by participants. A follow-up was done to confirm participation. Informed consent forms were signed by participants who agreed to take part. For those unable to read or write, forms were translated into their preferred local language in the presence of a witness, after which both the participant and witness signed or thumb-printed the form. Participants were informed that even after signing, they retained the right to withdraw without repercussions. Anonymity was guaranteed by assigning codes to participants in the order of recruitment (PA, PB, PC……PP). Collected information, including consent forms and recordings, is securely stored and only accessible to only the research team. The electronic files were stored in password-protected folders in the personal computer of the 1st author. All the participants consented for their interview transcripts to be published, and we ensured all personal identifiers were removed from the manuscript. In cases where participants became emotionally distressed while recounting their experiences, arrangements were made for professional counselling support.

## Data analysis

In this study, analysis was conducted alongside data collection. Each interview was analyzed before the next was conducted, allowing emerging insights to inform subsequent interviews and enhance the richness of the data. All interviews were audio-recorded and listened to several times to ensure familiarity and immersion in the data. Interviews conducted in Twi and Ga were translated and transcribed into English with the assistance of qualified language experts in Twi and Ga, ensuring accuracy, tone, and cultural meaning were maintained. The translated transcripts in English language were compared with the original Twi and Ga recordings to confirm that participants' intentions and emotions were preserved. The transcribed data were analyzed using the descriptive phenomenological method, which involves reading the transcripts to gain an overall sense of the data, identifying significant statements, transforming these statements into meaningful units, grouping similar meanings into clusters, and organizing them into overarching themes that capture the essence of participants' experiences. NVivo version 11 software was used to systematically manage and organize the codes to enhance transparency and rigor. Themes were generated around the constructs of the social cognitive theoretical framework illustrated in Fig 1 above.

## Results

The study involved sixteen postpartum mothers aged between 21 and 42 years, reflecting a broad range of reproductive experiences. In terms of parity, the majority had between two and four children, while a few were first-time mothers. At the time of the interviews, participants were between one week and seven months postpartum, representing varying stages of recovery and adjustment after childbirth. Most women shared that their hypertension was first detected during pregnancy,

whereas a few reported its onset after delivery. Their management primarily involved Nifecard (XL 30–90 mg) and Methyldopa, either taken singly or in combination, under the guidance of healthcare professionals. These medications formed an integral part of their daily routine as they navigated the demands of motherhood alongside managing their health. The background information is presented in Table 1.

The themes and sub-themes that were generated from the data is shown in Table 2.

The findings of this study revealed three major themes that encapsulated the multifaceted experiences of postpartum mothers living with hypertension in the Accra Metropolis. These themes, expectations, self-efficacy and environmental influences.

Each theme is discussed below, supported by participants' quotations to illustrate their lived realities, and interpreted through the lens of the Social Cognitive Theory (SCT).

## Expectations about living with hypertension

This theme reflects the perceptions and emotional outlook of postpartum mothers as they navigated life with hypertension following childbirth. Two sub-themes emerged: optimism and hope for recovery, and fear and uncertainty about complications. These expectations influenced how participants engaged in self-care, adhered to treatment, and adjusted psychologically. In line with Social Cognitive Theory (SCT), such expectations correspond to the construct of outcome expectancies, which guide motivation and participation in health-promoting behaviors [20,21]. Overall, mothers' expectations reflected a combination of optimism and apprehension. Some expressed hope and confidence in recovery, while others felt overwhelmed by the chronic nature of hypertension. According to SCT, these varying expectations influence behavior: positive outlooks encourage proactive health actions, whereas fear and uncertainty may lead to avoidance or emotional distress.

**Table 1. Background Information of Postpartum Mothers Living with Hypertension.**

| Participant | Age (years) | Marital Status | Number of Children | Time Since Delivery | Diagnosis Period | Current Treatment | Language Used |
|---|---|---|---|---|---|---|---|
| PA | 21 | Married | 1 | 3 weeks | During pregnancy | Nifecard 30 mg | English & Twi |
| PB | 24 | Married | 2 | 2 weeks | During pregnancy | Methyldopa 90 mg | Twi |
| PC | 27 | Married | 3 | 1 month | During pregnancy | Nifecard XL 30 mg | English |
| PD | 30 | Married | 2 | 3 weeks | After delivery | Methyldopa 125 mg | English & Twi |
| PE | 32 | Married | 3 | 2 weeks | During pregnancy | Nifecard XL 90 mg | English |
| PF | 28 | Married | 2 | 1 week | During pregnancy | Methyldopa & Neficard | Twi & Ga |
| PG | 29 | Married | 2 | 2 weeks | During pregnancy | Nifecard 30 mg daily | English & Ga |
| PH | 33 | Married | 4 | 3 weeks | During pregnancy | Nifecard XL 30 mg daily | English & Twi |
| PI | 31 | Married | 3 | 1 month | After delivery | Nifecard 30 mg daily | English |
| PJ | 35 | Married | 4 | 2 weeks | During pregnancy | Nurse-administered (unspecified) | Twi |
| PK | 42 | Married | 4 | 2 weeks | During pregnancy | Nifecard XL 90 mg | English & Twi |
| PL | 38 | Married | 3 | 1 week | During pregnancy | Nifecard XL 30 mg | English |
| PM | 42 | Married | 4 | 1 week | During pregnancy | Nifecard + Methyldopa | Twi |
| PN | 32 | Married | 1 | 2 weeks | After pregnancy | Nifecard 30 mg | English |
| PO | 34 | Married | 4 | 1 week | After pregnancy | Nifecard 90 mg | English |
| PP | 42 | Married | 1 | 5 weeks | During pregnancy | Nifecard XL 90 mg + Methyldopa | English |

**Table 2. Themes and sub-themes identified from the data.**

| Main Theme | Sub-Themes | Summary Description |
|---|---|---|
| Expectations about Living with Hypertension | 1. Optimism and hope for recovery2. Fear and uncertainty about complications | Participants expressed mixed feelings about their condition. While some held optimistic beliefs about recovery through medication, healthy lifestyles, and divine intervention, others worried about future complications or recurrence in subsequent pregnancies. These expectations influenced their motivation to adhere to care. |
| Self-Efficacy in Managing Hypertension | 1. Confidence in adhering to treatment2. Challenges with lifestyle modification and monitoring | Participants demonstrated varying degrees of confidence in managing their condition. Those with higher self-efficacy-maintained medication routines and dietary adjustments, while others struggled due to stress, limited resources, and lack of monitoring devices. Their perceived ability to manage hypertension reflected the core concept of self-efficacy within the Social Cognitive Theory, where belief in one's capacity determines persistence in health-related behavior. |
| Environmental factors | 1. Spousal and family support2. Role of healthcare providers3. Social perceptions and stigma | Support from spouses, relatives, and healthcare providers played a critical role in shaping mothers' adherence and emotional well-being. Positive support enhanced motivation, whereas stigma or insensitive attitudes from the community and some healthcare workers reduced confidence and engagement in care. |

## Optimism and hope for recovery

Several mothers expressed a positive perspective on recovery and effectively managing hypertension. They believed that adhering to medical advice, making lifestyle adjustments, and praying would help restore their health and allow them to fully resume maternal responsibilities. Their optimism reflects strong outcome expectations, a core SCT concept emphasizing belief in the benefits of desired behaviors. Some of the participants' quotes are below.

*"I know if I keep taking my medicine and reduce stress, my blood pressure will come down. I am getting better, and I believe by God's grace it will be normal again."* (Participant C)*"The doctor told me it will stabilize if I follow the routine, so I don't see it as a permanent sickness. I am doing everything they say."* (Participant F)*"Sometimes I just pray and trust God that I won't live with it forever. I am hopeful."* (Participant L)

Optimism strengthened their motivation to adhere to treatment and manage stress, consistent with SCT's emphasis on expecting positive health outcomes as a driver of self-regulation.

## Fear and uncertainty about complications

In contrast, some mothers reported anxiety and uncertainty about their health, describing hypertension as unpredictable and worrisome. Concerns centered on potential complications, recurrence in future pregnancies, and long-term dependence on medication. These feelings reflected low perceived control and reduced confidence, which can undermine self-efficacy and adherence to treatment. Some of the participants' quotes were:

*"Sometimes I worry because I don't know if this BP (blood pressure) will ever go away. What if it comes back when I get pregnant again?"* (Participant A)*"When they said my pressure was high again, I got scared. I thought it meant something was seriously wrong with me."* (Participant G)*"I almost collapsed before delivery because of BP (blood pressure), so anytime I feel dizzy, I panic. I think it will happen again."* (Participant I)

These accounts illustrate how negative outcome expectations can reduce motivation and coping. Mothers who doubted their ability to prevent or control complications often experienced higher emotional distress and hesitated to implement lifestyle changes.

### Self-efficacy in managing hypertension

This theme examines mothers' confidence in managing hypertension through medication adherence, lifestyle changes, and blood pressure monitoring. It reflects the self-efficacy construct of SCT, which emphasizes the belief in one's ability to perform behaviors that affect health outcomes [20,21]. Participants' experiences ranged from strong confidence in managing their condition to struggles caused by fatigue, stress, and limited resources. Two sub-themes were identified: confidence in treatment adherence and challenges with lifestyle modification and monitoring.

### Confidence in adhering to treatment

Many mothers reported confidence in following prescribed treatment routines. They consistently took medications, attended clinic appointments, and adhered to medical advice. This confidence was reinforced by information received during antenatal and post-natal care, as well as support from healthcare providers and family members. Some of the participants' quotes:

*"I never skip my medication because I know it keeps my blood pressure down. Even when I travel, I carry it along."* (Participant D)*"The nurse taught me how to take it at the same time every day. I am used to it now; it has become part of my routine."* (Participant H)

Faith and determination strengthened some women's confidence:

*"I tell myself I can manage it. I pray and take my drugs, and I see improvement."* (Participant J)*"When I was first diagnosed, I was scared. But now I am confident because I take the medicine and watch my diet. I know I can handle it."* (Participant N)

These accounts show that confidence grew through practice, education, and observing improvement—key aspects of self-efficacy according to SCT. Positive experiences reinforced mothers' belief in their ability to manage hypertension, fostering self-regulation and resilience.

### Challenges with lifestyle modification and monitoring

Despite their commitment, many mothers found it difficult to maintain lifestyle changes such as dietary restrictions, exercise, and regular blood pressure checks. Limited finances, household responsibilities, and lack of monitoring devices reduced their sense of control. These challenges illustrate SCT's assertion that environmental factors can limit health-promoting behaviors. Some of the participants' quotes:

*"Sometimes I forget to check my pressure because I don't have a machine at home, and going to the hospital all the time is costly."* (Participant B)*"They said I should avoid salty food, but it's not easy because we all eat the same food at home."* (Participant F)

Some highlighted competing demands:

*"With the baby and household chores, I hardly get time to rest or even think about what I am supposed to eat."* (Participant K)

Others noted emotional strain:

*"At times I feel tired and stressed, and that alone makes my pressure rise. I try to follow the advice, but it's not always possible."* (Participant M)

Postpartum mothers showed varying levels of confidence in managing hypertension. Those with strong self-efficacy consistently adhered to medications and engaged in positive health behaviors, whereas others struggled with lifestyle modifications due to stress, limited resources, and inadequate support. Consistent with SCT, self-efficacy is essential for effective hypertension management as it influences motivation, persistence, and adaptive coping.

### Environmental factors

This theme examines how external environmental factors, including spousal involvement, family support, relationships with healthcare providers, and social perceptions, shaped mothers' hypertension management. According to Social Cognitive Theory (SCT), health behaviors result from continuous interactions between personal factors, environmental conditions, and behavioral responses [20,21]. In this study, environmental influences acted as both enabling and constraining factors, affecting adherence, emotional well-being, and motivation. Environmental factors significantly shaped mothers' management of postpartum hypertension. Spousal and family support motivated adherence, and positive healthcare interactions reinforce self-efficacy. Conversely, stigma and negative social perceptions create emotional distress and secrecy. SCT underscores that behavior change and self-management are influenced by both social and environmental contexts.

Three sub-themes emerged: spousal and family support, role of healthcare providers and social perceptions and stigma.

### Spousal and family support

Many mothers acknowledged that support from spouses and family was essential to managing hypertension effectively. Encouragement, reminders to take medication, financial help, and emotional reassurance were critical for adherence and recovery. Supportive family relationships enhanced confidence and reduced stress, reinforcing positive health behaviors. Some of the participants' quotes were:

> *"My husband reminds me to take my medicine and even buys them for me when I forget."* (Participant F) *"My mother helps me with the baby so I can rest and take my medication. Without her, I would have broken down."* (Participant J) *"When my pressure was high, my husband was always by me, encouraging me and saying I would be fine."* (Participant C)

However, some mothers reported limited or absent support. One shared.

> *"I live alone with my child; there is no one to check on me or remind me to take my drugs."* (Participant A)

Another participant stated:

> *"Sometimes, you have no one to even depend on as your own husband desserts you."* (Participant D)

These accounts illustrate SCT's principle of reciprocal determinism, showing how supportive social environments enhance self-efficacy and health-promoting behaviors, while the lack of support can increase vulnerability and reduce motivation.

### Role of healthcare providers

Healthcare professionals, particularly nurses and midwives, played a central role in mothers' recovery. Participants valued guidance, emotional reassurance, and professional care during clinic visits. Positive interactions reinforced confidence in self-care and strengthened adherence, consistent with SCT's emphasis on modelling and reinforcement. Some of the participants' quotes were.

*"The nurses are patient with me. They check my blood pressure and explain what I should do to bring it down."* (Participant A) *"Sometimes I go there feeling weak, and they comfort me. They even share stories of others who are doing well to encourage me."* (Participant K) *"The midwife calls me when I miss my appointment and advises me not to stop my medication. That has helped me a lot."* (Participant B)

Some mothers, however, experienced limited attention due to staff workload:

*"Sometimes the nurses are too busy. They just check the pressure and give the drugs without much talk."* (Participant M)

These narratives highlight that consistent, empathetic healthcare interactions can strengthen self-efficacy, whereas insufficient support may undermine trust and engagement.

### Social perceptions and stigma

Some mothers reported encountering negative social judgments and stigma regarding their hypertension. Community members and even healthcare staff sometimes viewed them as "too young" to have high blood pressure, causing feelings of shame and isolation and discouraging openness about their condition. Below are some of the quotes from the participants.

*"When I went to the pharmacy, they said, 'Ah, small girl like you with blood pressure?' It made me feel bad."* (Participant P) *"Some people think only old people get high blood pressure, so when they see me taking medicine, they laugh."* (Participant D) *"I hide my drugs when friends visit because I don't want questions."* (Participant H)

These accounts show how stigma and misconceptions can disrupt self-regulation and reduce motivation. SCT highlights that negative environmental cues may hinder health-promoting behaviors, suggesting that public education could help normalize postpartum hypertension.

## Discussion

This study explored postpartum mothers' experiences regarding the management of hypertension. The findings revealed that the participants had diverse expectations of their diagnosis. Some were optimistic, expressing confidence on medication and lifestyle modifications as well as faith in divine interventions leading to recovery. Others experienced fear, anxiety, and uncertainty about complications or recurrence in future pregnancies. These expectations shaped their emotional well-being, motivation, and adherence to treatment. The findings point to the psychological implications of living with hypertension and this needs to be tackled to improve the quality of life of these women. According to Social Cognitive Theory, outcome expectations as well as beliefs about the likely results of one's actions play a critical role in health behaviors [19,21,22]. The study found that mothers who believed that consistent medication use, regular monitoring, and stress management would improve their health demonstrated greater commitment to self-care. Conversely, those who doubted the effectiveness of these measures or feared long-term complications showed lower confidence and reduced adherence. This finding is consistent with the assertions of previous studies [11,23,24] that stated that positive expectations foster persistence in managing chronic health conditions. These findings also align with prior research. For example, [25] reported that Ghanaian women with hypertensive disorders in pregnancy who anticipated positive outcomes were more likely to comply with postpartum care and medication. Similarly, Mensah and Obeng [18] found that women who believed their blood pressure could normalize after childbirth were more motivated to follow medical advice and adopt healthy behaviors. In contrast, fear of complications negatively influenced engagement in care, consistent with [10,25], who noted that

women perceiving hypertension as a lifelong condition often expressed frustration and hopelessness and thus reducing adherence. The findings of this study point to the need for health workers to institute measures towards building trust and confidence in the treatment being given to postpartum women with hypertension to promote better adherence to treatment.

Spirituality also played a significant role in shaping the experiences of the participants in this study. The participants relied on prayer and faith in God, interpreting their recovery as partly dependent on divine intervention. These supports [16,22,26–30] observation that Ghanaian mothers often integrate spirituality into health-seeking behaviors, using faith as a source of emotional strength when diagnosed with chronic diseases. Within the SCT, spiritual beliefs can serve as self-reinforcement, sustaining motivation and coping [30]. However, the coexistence of hope and fear illustrates the psychological complexity of adjusting to postpartum hypertension. Participants with low spiritual beliefs appeared uncertain about recurrence or long-term effects as compared with their counterparts. High spiritual belief often leads to increased confidence and general positive coping with the disease as echoed in other studies [22,26,27], who noted that prior experiences with women with low spiritual beliefs during pre-eclampsia heightened anxiety and reduced adherence to follow-up care. From a SCT perspective, low positive and confidence weakens self-efficacy and hinders proactive health behaviors [20,21]. The emergence of spirituality in the experiences of the participants as a tool for promoting confidence provides an emerging resource to help postpartum women cope and adjust to hypertension and this can be explored by nurses and midwives working in collaboration with religious leaders to promote treatment adherence.

Furthermore, the study found that social network of the participants played a crucial role in promoting self-confidence and acted as social reinforcement which promoted treatment adherence. The results from this study showed that the participants with higher self-efficacy described maintaining medication schedules, attending clinic appointments, and implementing dietary recommendations. Positive health outcomes, ongoing guidance from healthcare providers, and family encouragement reinforced their confidence, supporting [17,26] claim that mastery experiences and social reinforcement enhance self-regulation in health management. However, some of the participants reportedly struggled to sustain dietary restrictions, regular exercise, and blood pressure monitoring due to limited financial resources, caregiving responsibilities, and psychological fatigue [2,4,16]. These barriers align with [26–29] who reported that economic hardship and family obligations hindered postpartum women's self-care in Ghana. Similarly, Obeng et al. [14] found that low self-efficacy among women with chronic conditions was associated with inconsistent medication use and delayed health-seeking behavior. The findings of this study give a basis for nurses and midwives to involve the social networks of postpartum women diagnosed with hypertension in their management. These social networks, which often include immediate family members, friends and colleagues from religious networks could play a crucial role on social reinforcement of treatment.

In addition, supportive family environments were found to foster motivation and confidence, while isolation and stress undermined self-management as they increased the social stigma like those stated in other studies [13,26,28]. Emotional resilience built through supportive family environment serves as a resource which promotes treatment adherence. Some of the mothers reported that prayers with close family relatives and positive thinking from close relatives and friends helped them persevere with treatment despite challenges and helped them feel wanted hence reduced the stigma associated with their conditions. This aligns with Yeboah et al. [17], who observed that faith-based supportive networks reinforce self-efficacy by promoting hope and determination. Overall, self-efficacy was dynamic, evolving through experience, social reinforcement, and health outcomes. The participants claimed that their confidence increased with observable health improvements or affirmation from close friends, healthcare providers and family, while unmet expectations, stress, or lack of support diminished confidence and treatment adherence. These findings echo Mensah et al.'s work [18], who highlighted that self-efficacy develops through continuous learning and reinforcement in the social context. The findings collaborate with earlier studies [18,26,27] which emphasize the role of social networks in promoting resilience among similar populations and provide opportunities for targeted care.

Also, the study found that the participants who received encouragement, reminders, and practical assistance from spouses and family members were more likely to adhere to medications and maintain healthy routines. This aligns with [17,26,28], who observed that spousal and family support fosters self-efficacy, emotional well-being, and adherence among postpartum women managing chronic illnesses in Ghana. Evidence from Ghana also shows that access to supportive environments and self-monitoring tools improves mothers' confidence in managing hypertension [6,25,22]. Conversely, mothers lacking family or partner support reported isolation, emotional distress, and difficulty following medical advice, consistent with other studies [26,27], who noted that limited social support contributes to poor self-management and heightened anxiety. Similarly, healthcare providers who provided empathetic communication, education, and follow-up increased the participants' confidence and trust in treatment. Positive interactions reinforced adherence by providing guidance and reassurance, echoing Appiah et al. [19] work, who emphasized that supportive midwife-patient relationships enhanced treatment adherence and emotional resilience. Conversely, some participants reported rushed consultations and limited attention from overburdened staff, reducing satisfaction and motivation, reflecting health system constraints as highlighted by [31]. Nurses and midwives providing care for postpartum women diagnosed with hypertension must see spouses and close family members of such women as resources that can help the women cope with their diagnoses and adhere to treatment.

The study showed that social stigma and misconceptions about hypertension further influenced experiences of the participants. Several mothers reported ridicule or misunderstanding from peers who perceived hypertension as a disease of older adults. They claimed that the stigmatization led to secrecy, shame, and reluctance to seek help, supporting Boateng and Nartey [16], who noted that chronic illness stigma fosters emotional distress and avoidance behaviors. SCT suggests that such negative social cues disrupt self-regulation and reduce engagement in health-promoting behaviors. Overall, the findings indicate that the environment profoundly affects postpartum mothers' management of hypertension. Supportive relationships with family, spouses, peers, and healthcare providers fostered motivation, emotional stability, and adherence, while social stigma and healthcare system limitations undermined confidence and self-care. SCT underscores that health behaviors are shaped through interactions between individuals and their social environments. Strengthening supportive ecosystems through family education, community awareness, and improved healthcare engagement is crucial. Health policies should emphasize patient-centered care, spousal involvement, and stigma reduction to enhance self-efficacy, adherence, and overall well-being among postpartum women with hypertension.

## Limitations of the study

The study was hospital-based and did not involve other women who were not receiving care at the study settings. The findings cannot represent the views of all women diagnosed with hypertension during the postnatal period.

The data could have been affected by recall bias even though efforts were made to ensure the participants gave a vivid account during the study

Furthermore, the findings like most questionnaire-based studies could be affected by social desirability issues. However, the researchers asked the participants to give a vivid account of their experiences as per the interview guide and this reduced the social desirability issues.

## Conclusion and recommendations

Post-partum women diagnosed with hypertension face social stigma and have mixed feelings of hope, fear and uncertainty about complications. Effective management of postpartum hypertension requires patient-centered and holistic interventions that enhance women's self-efficacy, strengthen family and community support, reduces stigma and improve nurse/midwife–client communication. Family members and close friends as well as religious networks of postpartum women diagnosed with hypertension could promote confidence, emotional resilience and enhance treatment adherence.

It is recommended that the Ghana Health Service and managers of post-natal clinics must design targeted postpartum hypertension education programs aimed at reducing social stigma and integrates the role of spouses and social networks of postnatal women diagnosed with hypertension including faith-based organizations to improve compliance with treatment and lifestyle modifications

## Supporting information

**S1 File. Interview guide.**
(DOCX)

## Author contributions

**Conceptualization:** Kennedy Dodam Konlan, Cecilia Eliason, Hellen Akosua Asante.

**Investigation:** Cecilia Eliason.

**Methodology:** Kennedy Dodam Konlan, Cecilia Eliason, Hellen Akosua Asante.

**Project administration:** Hellen Akosua Asante.

**Supervision:** Kennedy Dodam Konlan, Cecilia Eliason.

**Writing – original draft:** Kennedy Dodam Konlan.

**Writing – review & editing:** Hellen Akosua Asante.

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
