## [Decision Letter · Decision Letter 0]

22 Dec 2025

PONE-D-25-62035Lived experiences of postpartum women diagnosed with hypertension receiving care in a resource-constrained setting in Accra, GhanaPLOS One

Dear Dr. Konlan,

Thank you for submitting your manuscript to PLOS ONE. After careful consideration, we feel that it has merit but does not fully meet PLOS ONE’s publication criteria as it currently stands. Therefore, we invite you to submit a revised version of the manuscript that addresses the points raised during the review process.

We look forward to receiving your revised manuscript.

Kind regards,

Joyce Jebet Cheptum

Academic Editor

PLOS One

**Journal Requirements:**

1. When submitting your revision, we need you to address these additional requirements. Please ensure that your manuscript meets PLOS ONE's style requirements, including those for file naming. The PLOS ONE style templates can be found at https://journals.plos.org/plosone/s/file?id=wjVg/PLOSOne_formatting_sample_main_body.pdf and https://journals.plos.org/plosone/s/file?id=ba62/PLOSOne_formatting_sample_title_authors_affiliations.pdf 2. Your ethics statement should only appear in the Methods section of your manuscript. If your ethics statement is written in any section besides the Methods, please move it to the Methods section and delete it from any other section. Please ensure that your ethics statement is included in your manuscript, as the ethics statement entered into the online submission form will not be published alongside your manuscript. 3. We note that this data set consists of interview transcripts. Can you please confirm that all participants gave consent for interview transcript to be published? If they DID provide consent for these transcripts to be published, please also confirm that the transcripts do not contain any potentially identifying information (or let us know if the participants consented to having their personal details published and made publicly available). We consider the following details to be identifying information:- Names, nicknames, and initials- Age more specific than round numbers- GPS coordinates, physical addresses, IP addresses, email addresses- Information in small sample sizes (e.g. 40 students from X class in X year at X university)- Specific dates (e.g. visit dates, interview dates)- ID numbers Or, if the participants DID NOT provide consent for these transcripts to be published:- Provide a de-identified version of the data or excerpts of interview responses- Provide information regarding how these transcripts can be accessed by researchers who meet the criteria for access to confidential data, including:a) the grounds for restrictionb) the name of the ethics committee, Institutional Review Board, or third-party organization that is imposing sharing restrictions on the datac) a non-author, institutional point of contact that is able to field data access queries, in the interest of maintaining long-term data accessibility.d) Any relevant data set names, URLs, DOIs, etc. that an independent researcher would need in order to request your minimal data set. For further information on sharing data that contains sensitive participant information, please see: https://journals.plos.org/plosone/s/data-availability#loc-human-research-participant-data-and-other-sensitive-data If there are ethical, legal, or third-party restrictions upon your dataset, you must provide all of the following details (https://journals.plos.org/plosone/s/data-availability#loc-acceptable-data-access-restrictions):a) A complete description of the datasetb) The nature of the restrictions upon the data (ethical, legal, or owned by a third party) and the reasoning behind themc) The full name of the body imposing the restrictions upon your dataset (ethics committee, institution, data access committee, etc)d) If the data are owned by a third party, confirmation of whether the authors received any special privileges in accessing the data that other researchers would not havee) Direct, non-author contact information (preferably email) for the body imposing the restrictions upon the data, to which data access requests can be sent 4. We note that there is identifying data in Table 1. Due to the inclusion of these potentially identifying data. Prior to sharing human research participant data, authors should consult with an ethics committee to ensure data are shared in accordance with participant consent and all applicable local laws. Data sharing should never compromise participant privacy. It is therefore not appropriate to publicly share personally identifiable data on human research participants. The following are examples of data that should not be shared: -Name, initials, physical address-Ages more specific than whole numbers-Internet protocol (IP) address-Specific dates (birth dates, death dates, examination dates, etc.)-Contact information such as phone number or email address-Location data-ID numbers that seem specific (long numbers, include initials, titled “Hospital ID”) rather than random (small numbers in numerical order) Data that are not directly identifying may also be inappropriate to share, as in combination they can become identifying. For example, data collected from a small group of participants, vulnerable populations, or private groups should not be shared if they involve indirect identifiers (such as sex, ethnicity, location, etc.) that may risk the identification of study participants. Additional guidance on preparing raw data for publication can be found in our Data Policy (https://journals.plos.org/plosone/s/data-availability#loc-human-research-participant-data-and-other-sensitive-data) and in the following article: http://www.bmj.com/content/340/bmj.c181.long. Please remove or anonymize all personal information (<specific identifying information in file to be removed>), ensure that the data shared are in accordance with participant consent, and re-upload a fully anonymized data set. Please note that spreadsheet columns with personal information must be removed and not hidden as all hidden columns will appear in the published file. 5. If the reviewer comments include a recommendation to cite specific previously published works, please review and evaluate these publications to determine whether they are relevant and should be cited. There is no requirement to cite these works unless the editor has indicated otherwise.

**Additional Editor Comments:**

Add "resource-constrained setting" in the keywords.

The conclusions need to be more succinct and be related to the study title and findings. The current conclusion is based on the management.

Reviewers' comments:

Reviewer's Responses to Questions

**Comments to the Author**

1. Is the manuscript technically sound, and do the data support the conclusions?

Reviewer #1: Yes

Reviewer #2: Yes

2. Has the statistical analysis been performed appropriately and rigorously? 

Reviewer #1: I Don't Know

Reviewer #2: No

3. Have the authors made all data underlying the findings in their manuscript fully available?

Reviewer #1: Yes

Reviewer #2: Yes

4. Is the manuscript presented in an intelligible fashion and written in standard English?

Reviewer #1: Yes

Reviewer #2: Yes

5. Review Comments to the Author

**Reviewer #1:** Manuscript Number: PONE-D-25-62035

This manuscript investigates the lived experiences of postpartum women diagnosed with hypertension receiving care in a resource-constrained setting in Accra, Ghana, using a qualitative descriptive phenomenological design based on the social cognitive theory. The author collected data from 16 women using semi-structured in-depth interviews. The manuscript content demonstrates scientific rigor for a qualitative study. The manuscript is written in English but there are minor grammatical errors. For example the drug 'Nifecard' is written as 'Nefecard'

Background

• The current background emphasizes the epidemiology and burden of postpartum hypertension but does not sufficiently address the lived experiences of affected women. Strengthening the background to reflect existing evidence on women’s experiences and clearly identifying the gap your study intends to fill will enhance the rationale.

• The theoretical framework is not described before its application in the study. Describe the constructs and how they informed the thematic analysis.

Methods

• The study design is appropriate, and the sampling is well described, including how data saturation was achieved.

• The study design description appears incomplete (check the paragraph under study design the last sentence is floating).

• In the inclusion criteria, describe how you will assess to confirm a cognitively stable mother. Did you have a cutoff for the age of the mother?

• The authors have not included the NVivo data codes or codebook used in the analysis. Without this information, the transparency of the analytic process is limited, and it is unclear which themes were most recurrent or how the relative weight of each theme was determined.

Results

• The results are well captured with participants' demographics clearly displayed in a table, and participants' excerpts are also indicated as appropriate for each subtheme/theme.

• A participant who was 7 months postpartum appears inconsistent with the rest of the sample, as the other mothers were 1 week to 1 month postpartum. Can this participant be considered postpartum at this time?

Discussion and Conclusions

• The comparison with other studies is noted, but the author’s independent scholarly contribution is not clear. The discussion should more explicitly articulate the significance of the findings and provide a clear response to the “so what” question.

• In the discussion, the study design changes to cross-sectional instead of descriptive phenomenology

• The discussion mentions that the study explored the expectations of the postpartum mothers instead of the experiences.

• The limitations are well articulated.

• The conclusion may need strengthening. Conclude based on the study findings

Ethical Considerations: The ethical consideration details are well captured and complete.

**Reviewer #2:**Title: I recommend rephrasing it for better clarity, propose "Lived experiences of postpartum hypertensive women in Accra, Ghana." Other waste words put in the title should be explained in the methodology and setting.

Abstract: It has almost the recommended word count- 305. It however, has some typo errors i.e. under method- purpose corrected to purposively; with the aid NVivo corrected to with the aid of NVivo.

On the main manuscript, under study design- last sentence ending "through" seems incomplete.

Selection of participants and data collection- paragraph 2, sentence 4: correct to "by the research team' instead of "researcher team". Sentence 7 (last sentence of paragraph 2): ? meaning of sentence i.e. "to grant the interviews". Suggest revision of the sentence.

Participants are 16 and researcher coded them using the alphabet and one would expect the coding to end at letter P but quotes from participants are given up to participant S. Need to correct this.

Intext citation seems to be mixing up various formats i.e. writing some author names as well as numbering them again.

6. PLOS authors have the option to publish the peer review history of their article (what does this mean?). If published, this will include your full peer review and any attached files.

Reviewer #1: **Yes:**Dr. Serah Wanjiru Wachira

Reviewer #2: **Yes:**Dr Grace Danda

---

## [Author Response · Author response to Decision Letter 1]

25 Dec 2025

University of Ghana

College of Health Sciences

School of Nursing and Midwifery

Department of Adult Health

25th DECEMBER, 2025

The Editor

PLOS ONE

Dear Sir/Madam,

Response to review

General comments of authors

We have addressed all the comments of the editor and reviewers as suggested and we hope the revised manuscript meets the standards for publications.

Comments of the Editor

PONE-D-25-62035

Lived experiences of postpartum women diagnosed with hypertension receiving care in a resource-constrained setting in Accra, Ghana

PLOS One

Dear Dr. Konlan,

Thank you for submitting your manuscript to PLOS ONE. After careful consideration, we feel that it has merit but does not fully meet PLOS ONE’s publication criteria as it currently stands. Therefore, we invite you to submit a revised version of the manuscript that addresses the points raised during the review process.

• A letter that responds to each point raised by the academic editor and reviewer(s). You should upload this letter as a separate file labeled 'Response to Reviewers'.

We look forward to receiving your revised manuscript.

Kind regards,

Joyce Jebet Cheptum

Academic Editor

PLOS One

Journal Requirements:

3. We note that this data set consists of interview transcripts. Can you please confirm that all participants gave consent for interview transcript to be published?

If they DID provide consent for these transcripts to be published, please also confirm that the transcripts do not contain any potentially identifying information (or let us know if the participants consented to having their personal details published and made publicly available). We consider the following details to be identifying information:

- Names, nicknames, and initials

- Age more specific than round numbers

- GPS coordinates, physical addresses, IP addresses, email addresses

- Information in small sample sizes (e.g. 40 students from X class in X year at X university)

- Specific dates (e.g. visit dates, interview dates)

- ID numbers

Or, if the participants DID NOT provide consent for these transcripts to be published:

- Provide a de-identified version of the data or excerpts of interview responses

- Provide information regarding how these transcripts can be accessed by researchers who meet the criteria for access to confidential data, including:

a) the grounds for restriction

b) the name of the ethics committee, Institutional Review Board, or third-party organization that is imposing sharing restrictions on the data

c) a non-author, institutional point of contact that is able to field data access queries, in the interest of maintaining long-term data accessibility.

d) Any relevant data set names, URLs, DOIs, etc. that an independent researcher would need in order to request your minimal data set.

For further information on sharing data that contains sensitive participant information, please see: https://journals.plos.org/plosone/s/data-availability#loc-human-research-participant-data-and-other-sensitive-data

If there are ethical, legal, or third-party restrictions upon your dataset, you must provide all of the following details (https://journals.plos.org/plosone/s/data-availability#loc-acceptable-data-access-restrictions):

a) A complete description of the dataset

b) The nature of the restrictions upon the data (ethical, legal, or owned by a third party) and the reasoning behind them

c) The full name of the body imposing the restrictions upon your dataset (ethics committee, institution, data access committee, etc)

d) If the data are owned by a third party, confirmation of whether the authors received any special privileges in accessing the data that other researchers would not have

e) Direct, non-author contact information (preferably email) for the body imposing the restrictions upon the data, to which data access requests can be sent

4. We note that there is identifying data in Table 1. Due to the inclusion of these potentially identifying data. Prior to sharing human research participant data, authors should consult with an ethics committee to ensure data are shared in accordance with participant consent and all applicable local laws.

-Location data

Additional Editor Comments:

Add "resource-constrained setting" in the keywords.

The conclusions need to be more succinct and be related to the study title and findings. The current conclusion is based on the management.

Authors’ Response to Comments of editor

General Response to editor’s comments

We are grateful for the comments of the editor and have addressed all the concerns and comments of the editor.

Authors’ response to Journal Requirements:

1. We have ensured that our manuscript meets PLOS ONE's style requirements, including those for file naming.

2. We have ensured that our ethics statement appeared only in the Methods section of our manuscript. This is found on pages 10 and 11 of the revised manuscript.

3. We have stated on page 11 that all the participants gave consent for interview transcript to be published.

4. The information contained in Table 1 (pages 12 and 13) do not breach any ethical principle and does not disclose any information that could lead to identification of the participants. We have removed all personal identification information.

5. This was not applicable to our manuscript.

6. We have ensured all works cited are contained in the reference list as suggested. This is found on pages 23-27 of the revised manuscript.

Additional Editor Comments:

We have added "resource-constrained setting" in the keywords. This is found on page 2 of the revised manuscript.

We have re-written the conclusion to make it succinct and related to the study title and findings. This is found on page 2 (Abstract) and page 23 of the revised manuscript.

REVIEWER 1 COMMENTS

Reviewer #1: Manuscript Number: PONE-D-25-62035

This manuscript investigates the lived experiences of postpartum women diagnosed with hypertension receiving care in a resource-constrained setting in Accra, Ghana, using a qualitative descriptive phenomenological design based on the social cognitive theory. The author collected data from 16 women using semi-structured in-depth interviews. The manuscript content demonstrates scientific rigor for a qualitative study. The manuscript is written in English but there are minor grammatical errors. For example the drug 'Nifecard' is written as 'Nefecard'

Background

• The current background emphasizes the epidemiology and burden of postpartum hypertension but does not sufficiently address the lived experiences of affected women. Strengthening the background to reflect existing evidence on women’s experiences and clearly identifying the gap your study intends to fill will enhance the rationale.

• The theoretical framework is not described before its application in the study. Describe the constructs and how they informed the thematic analysis.

Methods

• The study design is appropriate, and the sampling is well described, including how data saturation was achieved.

• The study design description appears incomplete (check the paragraph under study design the last sentence is floating).

• In the inclusion criteria, describe how you will assess to confirm a cognitively stable mother. Did you have a cutoff for the age of the mother?

• The authors have not included the NVivo data codes or codebook used in the analysis. Without this information, the transparency of the analytic process is limited, and it is unclear which themes were most recurrent or how the relative weight of each theme was determined.

Results

• The results are well captured with participants' demographics clearly displayed in a table, and participants' excerpts are also indicated as appropriate for each subtheme/theme.

• A participant who was 7 months postpartum appears inconsistent with the rest of the sample, as the other mothers were 1 week to 1 month postpartum. Can this participant be considered postpartum at this time?

Discussion and Conclusions

• The comparison with other studies is noted, but the author’s independent scholarly contribution is not clear. The discussion should more explicitly articulate the significance of the findings and provide a clear response to the “so what” question.

• In the discussion, the study design changes to cross-sectional instead of descriptive phenomenology

• The discussion mentions that the study explored the expectations of the postpartum mothers instead of the experiences.

• The limitations are well articulated.

• The conclusion may need strengthening. Conclude based on the study findings

Ethical Considerations: The ethical consideration details are well captured and complete.

Authors’ response to reviewer 1

We are grateful for the comments of the reviewer. We have addressed the minor grammatical errors and worked on the spelling of the drug 'Nifecard'. This is found on pages 12 and 13 as well as throughout the entire revised manuscript.

Background

• We have identified the gap our study sought to fill in the background of the revised manuscript. This is found on 4 of the revised manuscript.

• We have included the theoretical framework underpinning the study. This is found on pages 4-6 of the revised manuscript as suggested by the reviewer.

Methods

• We have ensured that study design is complete and the last sentence which appeared floating has been completed. This is found on page 7 of the revised manuscript.

• We have stated how we assessed and confirmed a cognitively stable mother and this is found on page 8 of the revised manuscript at the inclusion criteria.

We have stated cutoff for the ages of the mothers on page 8 of the revised manuscript.

• We have re-organized the data analysis section and this is found on page 11 of the revised manuscript.

Results

• We have worked on the error in Table 1 regarding the last participant. This is found on page 13 of the revised manuscript.

Discussion and Conclusions

• We have stated our independent scholarly contribution in the discussion and ensured the discussion is properly written. This is found on pages 19-23 of the revised manuscript.

• We have corrected the errors in the discussion and ensured that the conclusion on page 23 of the revised manuscript is well written and strengthened as suggested by the reviewer.

It is our hope that the revised manuscript meets the standards of the reviewer and the journal.

Reviewer 2 Report

Reviewer #2: Title: I recommend rephrasing it for better clarity, propose "Lived experiences of postpartum hypertensive women in Accra, Ghana." Other waste words put in the title should be explained in the methodology and setting.

Abstract: It has almost the recommended word count- 305. It however, has some typo errors i.e. under method- purpose corrected to purposively; with the aid NVivo corrected to with the aid of NVivo.

On the main manuscript, under study design- last sentence ending "through" seems incomplete.

Selection of participants and data collection- paragraph 2, sentence 4: correct to "by the research team' instead of "researcher team". Sentence 7 (last sentence of paragraph 2

---

## [Decision Letter · Decision Letter 1]

13 Feb 2026

PONE-D-25-62035R1Lived experiences of postpartum hypertensive women in Accra, Ghana.PLOS One

Dear Dr. Konlan,

Thank you for submitting your manuscript to PLOS ONE. After careful consideration, we feel that it has merit but does not fully meet PLOS ONE’s publication criteria as it currently stands. Therefore, we invite you to submit a revised version of the manuscript that addresses the points raised during the review process.

We look forward to receiving your revised manuscript.

Kind regards,

Joyce Jebet Cheptum

Academic Editor

PLOS One

Journal Requirements:

Reviewer's Responses to Questions

**Comments to the Author**

1. If the authors have adequately addressed your comments raised in a previous round of review and you feel that this manuscript is now acceptable for publication, you may indicate that here to bypass the “Comments to the Author” section, enter your conflict of interest statement in the “Confidential to Editor” section, and submit your "Accept" recommendation.

Reviewer #1: All comments have been addressed

Reviewer #2: All comments have been addressed

2. Is the manuscript technically sound, and do the data support the conclusions?

Reviewer #1: Yes

Reviewer #2: Yes

3. Has the statistical analysis been performed appropriately and rigorously? 

Reviewer #1: Yes

Reviewer #2: Yes

4. Have the authors made all data underlying the findings in their manuscript fully available?

Reviewer #1: Yes

Reviewer #2: Yes

5. Is the manuscript presented in an intelligible fashion and written in standard English?

Reviewer #1: Yes

Reviewer #2: Yes

6. Review Comments to the Author

Reviewer #1: Reviewer comments

The authors have adequately addressed all the concerns raised in my initial review. The revisions have strengthened the manuscript, particularly in terms of clarity and methodological description. I have no major concerns and support the publication of the manuscript, subject to the following minor revisions:

Minor edits:

1. The abstract exceeds the recommended word limit (approximately 362 words). Please revise the conclusion and recommendations to be more concise.

2. Please clearly specify the source of Figure 1 (the theoretical framework), indicating whether it was developed by the authors, adapted, or adopted from existing literature, and provide an appropriate reference where applicable.

Reviewer #2: The authors have attended to all recommendations and the article is up to standard. I recommend acceptance of the manuscript.

7. PLOS authors have the option to publish the peer review history of their article (what does this mean?). If published, this will include your full peer review and any attached files.

Reviewer #1: **Yes:**Serah Wanjiru Wachira

Reviewer #2: **Yes:**Dr Grace Danda

---

## [Author Response · Author response to Decision Letter 2]

27 Mar 2026

University of Ghana

College of Health Sciences

School of Nursing and Midwifery

Department of Adult Health

26th March, 2026

The Editor

PLOS ONE

Dear Sir/Madam,

Response to review

General comments of authors

We have addressed all the comments of the reviewers as suggested and we hope the revised manuscript meets the standards for publications.

Reviewer #1:

Reviewer #1: Reviewer comments

The authors have adequately addressed all the concerns raised in my initial review. The revisions have strengthened the manuscript, particularly in terms of clarity and methodological description. I have no major concerns and support the publication of the manuscript, subject to the following minor revisions:

Minor edits:

1. The abstract exceeds the recommended word limit (approximately 362 words). Please revise the conclusion and recommendations to be more concise.

2. Please clearly specify the source of Figure 1 (the theoretical framework), indicating whether it was developed by the authors, adapted, or adopted from existing literature, and provide an appropriate reference where applicable.

Authors’ response to reviewer 1

We are grateful for the comments of the reviewer and have addressed all the comments in the revised manuscript as specifically indicated below

Minor edits:

1. We have worked on the abstract on page 2 of the revised manuscript to make same concise as recommended by the reviewer

2. We have specified that Figure 1 (the theoretical framework) was adopted and provided the appropriate citation. This is found on page 6 of the revised manuscript.

Reviewer #2:

The authors have attended to all recommendations and the article is up to standard. I recommend acceptance of the manuscript.

Authors’ response to reviewer 2

We are grateful for the reviewer’s comment of recommending acceptance of the manuscript and we have addressed the minor comments of the other reviewer.

We hope the revised manuscript will meet the standards for publication.

Thank you

Yours sincerely,

Dr. Kennedy Dodam Konlan

(Corresponding author)

---

## [Decision Letter · Decision Letter 2]

5 May 2026

Lived experiences of postpartum hypertensive women in Accra, Ghana.

PONE-D-25-62035R2

Dear Dr. Konlan,

We’re pleased to inform you that your manuscript has been judged scientifically suitable for publication and will be formally accepted for publication once it meets all outstanding technical requirements.

Kind regards,

Joyce Jebet Cheptum

Academic Editor

PLOS One

Additional Editor Comments (optional):

Reviewers' comments:

Reviewer's Responses to Questions

**Comments to the Author**

1. If the authors have adequately addressed your comments raised in a previous round of review and you feel that this manuscript is now acceptable for publication, you may indicate that here to bypass the “Comments to the Author” section, enter your conflict of interest statement in the “Confidential to Editor” section, and submit your "Accept" recommendation.

Reviewer #1: All comments have been addressed

Reviewer #2: All comments have been addressed

2. Is the manuscript technically sound, and do the data support the conclusions?

Reviewer #1: Yes

Reviewer #2: (No Response)

3. Has the statistical analysis been performed appropriately and rigorously? 

Reviewer #1: Yes

Reviewer #2: (No Response)

4. Have the authors made all data underlying the findings in their manuscript fully available?

Reviewer #1: Yes

Reviewer #2: (No Response)

5. Is the manuscript presented in an intelligible fashion and written in standard English?

Reviewer #1: Yes

Reviewer #2: (No Response)

6. Review Comments to the Author

Reviewer #1: I have thoroughly reviewed the revised manuscript and the authors’ responses to the reviewers’ comments. I am confident that the authors have sufficiently addressed all the concerns raised in the previous review.

In its current form, the manuscript meets the standards for publication. I therefore recommend that it be accepted for publication in PLOS ONE.

Reviewer #2: (No Response)

7. PLOS authors have the option to publish the peer review history of their article (what does this mean?). If published, this will include your full peer review and any attached files.

Reviewer #1: **Yes:**Serah Wanjiru Wachira

Reviewer #2: **Yes:**Grace Danda

---

## [Editor Report · Acceptance letter]

PONE-D-25-62035R2

PLOS One

Dear Dr. Konlan,

I'm pleased to inform you that your manuscript has been deemed suitable for publication in PLOS One. Congratulations! Your manuscript is now being handed over to our production team.

Kind regards,

on behalf of

Dr. Joyce Jebet Cheptum

Academic Editor

PLOS One